# Implementation of the Critical Thinking Blended Apprenticeship Curricula and Findings per Discipline: Foreign Language Teaching

**Ovidiu Ivancu** *[ID], **Roma Kriaučiūnienė** *[ID] **and Svetozar Poštič** *[ID]

Institute of Foreign Languages, Faculty of Philology, Vilnius University, 01513 Vilnius, Lithuania
* Correspondence: ovidiu.ivancu@flf.vu.lt (O.I.); roma.kriauciuniene@flf.vu.lt (R.K.);
  svetozar.postic@flf.vu.lt (S.P.)

**Abstract:** Critical thinking is a central element in higher education, designed to respond to authentic challenges that our society currently faces: the emergence and spread of fake news, disinformation, and manipulation. There is a consensus regarding CT's importance and role in higher education. Nevertheless, CT skills are often implicitly mentioned and only occasionally included in university curricula explicitly. The present paper aims at analysing how CT can be measured and tested in higher education, and it proposes specific tasks designed to increase the use of CT within the theoretical framework defined by Peter Facione and Paul Elder. Updated course descriptions were tested, and students' feedback was analysed and discussed. CT was measured by pre-questionnaires, mid-questionnaires, and post-questionnaires to establish the effectiveness of coherently implementing CT into the course descriptions. The survey includes answers to open questions to determine the suitability of the tasks proposed. The present research is part of the international Erasmus project "Critical Thinking for Successful Jobs".

**Keywords:** critical thinking; higher education; blended curriculum

## 1. Relevance of the Topic

There is an increasing need for critical thinking (CT) in higher education. Nevertheless, it is still problematic to achieve a consensus regarding a profile of a critical thinker. The debate is still open as to the best way to integrate CT into universities' curricula, since the process involves interdisciplinarity and transdisciplinarity [1].

On the one hand, we are all bombarded with information; on the other hand, part of the information made available to us may be the product of misinformation, manipulation, and propaganda. In the context of high competitiveness in the field of higher education, universities are compelled to provide attractive and efficient curricula and modern and active methods to increase the quality of teaching [2].

Given the context, the ability to organise, analyse, and interpret information is a vital skill that has to be cultivated in the academic environment. Thus, critical thinking may very well be a key concept in any intellectual process fostered by higher education institutions. Fostering CT is beneficial for students and educators, since it promotes authentic dialogues and facilitates the mediation between different worldviews [3].

Recently, in higher education there is a tendency to redesign curricula, shifting the focus from the subject taught to authentic tasks and real-life scenarios with the explicit purpose to enhance CT skills. Studies demonstrate that students and educators might have different perceptions of the outcome of CT; while educators are concerned with the process, students focus on the results [4].

"For many reasons, educators have become very interested in teaching *thinking skills* of various kinds in contrast with teaching information and content. Of course, you can do both, but in the past, the emphasis in most people's teaching has been on teaching

content [ . . . ] and though many teachers would claim to teach their teachers *how to think*, most would say that they do this indirectly or implicitly in the course of teaching the content which belongs to their special subject." [5].

In a study conducted by Aliakbari and Sadeghdaghighi [6], it was reported that teachers perceive three main interferences in the process of teaching CT: "The highest barrier was related to student characteristics. Self-efficacy ( . . . ) was the second main obstacle. Respondents reported lack of knowledge of the concept of critical thinking as the third high barrier to the implementation of critical thinking teaching strategies".

In the light of the recent changes in society (the increasing need for openness and social inclusion and the spread of fake news and disinformation), critical openness [7] constitutes a primary goal of modern education. Empirically, the presence of CT could be perceived as being intrinsic to the process of education. Indeed, the path to knowledge employs all the skills and dispositions of CT [8]. Nonetheless, there is still a lack of clear procedures.

The major challenge in teaching CT is the shift from implicit teaching to explicit teaching. The difficulty arises from the highly diversified taxonomy of CT [9]. Explicit teaching of CT implies that the educators are familiar with the theoretical framework of the concept and with the various models of implementation of CT in higher education [10].

Such a change requires an extensive set of teaching strategies and assessment grids. Currently, various researchers are trying to provide higher education institutions with specific toolkits, adjusted for the distinct needs of the subject taught [11].

According to a study published in 2008 [12], the explicit mention of CT in the curricula has a major impact on developing students' CT skills. The study suggests that for better results, CT skills should be developed separately and then integrated into the content of the course. For that purpose, teachers are required to have proper and professional training regarding CT.

A proper conceptual understanding of critical thinking should start with a brief history of its various meanings, the definition of the concept, and an accurate understanding of how it could be applied and integrated into the education system. Understanding the theoretical framework of CT might pave the way for a more comprehensive curriculum design in higher education. Transitioning from subject-centred curricula to learner-centred/problem-centred curricula implies encapsulating CT skills and dispositions adequately assessed and conveyed.

## 2. Introduction

The first historically attested form of critical thinking manifested itself in Ancient Greece, under the form of "Socratic questioning" or "Socratic dialogue" [13]. Socrates's method of teaching consisted of a series of carefully directed questions aimed at raising awareness. The goal was to distinguish between commonly accepted and appealing beliefs and ideas based solely on rhetoric and thoughts that could minimally pass the test of logic and clarity. The Socratic tradition was continued by many distinguished philosophers and thinkers of the Antiquity and Renaissance (Plato, Thomas Aquinas, Erasmus of Rotterdam, Thomas More, etc.).

In 1619, René Descartes started working on a treaty meant to establish scientific thinking methods. Although unfinished, the work lays a solid foundation for the critical thinking process. The twelve rules established by the French thinker in "Rules for the Direction of the Mind" remain to this day a cornerstone of critical thinking. One of them almost sounded like a prophecy: "There is a need for a method for finding out the truth" [14]. Descartes touches upon a fundamental challenge in the field of CT: the need for well-established theoretical frameworks.

In the 18th century, the French Enlightenment stressed the intrinsic connection between reason and valid thought processes. Montesquieu, Voltaire, and Diderot insisted on the idea that thinking requires rigour and discipline, ergo a method. The concept of critical thinking was widely and implicitly incorporated into various other fields, starting with the 18th century. Adam Smith applied it in the economy, Sigmund Freud and C.G Jung in

psychology, Darwin in biology, etc. The transversal use and value of the concept prove at the same time its importance and its volatility.

Only the 20th century registered a more explicit usage of the concept. William Graham Sumner advocates for criticism and scrutiny of ideas and beliefs as effective means of education in his extensive study "Folkways. A study of the sociological importance of usages, manners, customs, mores and morals":

> "The critical faculty is a product of education and training. It is a mental habit and power. It is a prime condition of human welfare that men and women should be trained in it. It is our only guarantee against delusion, deception, superstition, and misapprehension of ourselves and our earthly circumstances. It is a faculty which will protect us against all harmful suggestion." [15]

Abrami et al. (2014) suggest that the development of CT skills in HE is enhanced by "two general types of instructional interventions" [16]. On the one hand, the pedagogical context for dialogue and, on the other hand, the constant exposure of the student to authentic problems. Both are constitutive elements of the curriculum tested by us.

The present research aims at testing a newly designed blended curriculum, with the purpose of analysing the presence of CT skills within the academic environment of Vilnius University by applying the model described by Peter Facione. The model proposes a set of CT cognitive skills and sub-skills, which we intend to examine in our research (Table 1).

**Table 1.** CT cognitive skills and sub-skills [17].

| CT Cognitive Skills | Interpretation | Analysis | Evaluation | Inference | Explanation | Self-Regulation |
|---|---|---|---|---|---|---|
| CT cognitive sub-skills | Categorization Decoding significance Clarifying meaning | Examining ideas Identifying arguments Analyzing arguments | Assessing claims Assessing arguments | Querying evidence Conjecturing alternatives Drawing conclusions | Stating results Justifying procedures Presenting arguments | Self-examination Self-correction |

Assessing CT coherently and systematically is one of the most challenging tasks [18]. A problematic question immediately arises: can CT be evaluated using a scale universally applicable in all academic fields? CT requires certain dispositions (Table 2) that can be nurtured and identified transversally, regardless of the different disciplines studied in the educational environment. Peter Facione and Carlo Giancarlo propose a scale of "seven CT attribute-of-mind" [19].

**Table 2.** Critical thinking dispositions [19].

| | |
|---|---|
| Inquisitiveness | "one's intellectual curiosity and one's desire for learning even when the application of the knowledge is not readily apparent." |
| Systematicity | "being organized, orderly, focused, and diligent in inquiry." |
| Analyticity | "prizing the application of reasoning and the use of evidence to resolve problems, anticipating potential conceptual or practical difficulties, and consistently being alert to the need to intervene." |
| Truth seeking | "being eager to seek the best knowledge in a given context, courageous about asking questions, and honest and objective about pursuing inquiry even if the findings do not support one's self-interests or one's preconceived opinions." |
| Open-mindedness | "being tolerant of divergent views with sensitivity to the possibility of one's own bias." |

**Table 2.** *Cont.*

| | |
|---|---|
| CT self-confidence | "CT self-confidence allows one to trust the soundness of one's judgments and to lead others in the resolution of problems." |
| Maturity | "The CT- mature person can be characterised as one who approaches problems, inquiry, and decision making with a sense that some problems are necessarily ill-structured, some situations admit of more than one plausible option, and many times judgments must be made based on standards, contexts and evidence which preclude certainty." |

A well-trained critical thinker develops intellectual habits that require continuous training. The end goal is to enhance the ability to reason and understand. At the same time, CT can be perceived as an independent set of intellectual tools applicable both in a specialised field and in a broader context. The nurturing of such skills, as identified and defined by Facione [16], results in cultivating "universal intellectual traits" (Table 3).

**Table 3.** Universal intellectual traits [20].

| Intellectual Traits | Descriptors |
|---|---|
| Intellectual humility | To know that knowledge and research have limitations. |
| Intellectual autonomy | To think independently. |
| Intellectual integrity | To acknowledge the intellectual efforts of others. |
| Intellectual courage | To be ready to defend and present ideas against which people may manifest prejudices or biases. |
| Intellectual perseverance | To persist and be consistent and laborious. |
| Confidence in reason | To trust that reflective thinking is effective in problem-solving and decision-making processes. |
| Intellectual empathy | To put yourself in someone else's shoes, to reason, think, and view things from others' perspective. |
| Fairmindedness | To be impartial and objective. |

In the CT assessment, there is an increasing need to apply standards to measure and quantify scientifically the presence of the various skills and dispositions mentioned above. The empirical observation, though necessary from time to time, does not carry much scientific weight, and its findings are endangered by the ineluctable subjectivity of the observer. The process of fostering CT cognitive skills and sub-skills through pedagogic interventions (course description, lecture, workshop, seminars, etc.) ultimately develops intellectual standards that define a critical thinker.

Each of the intellectual standards (Table 4) defined by Paul& Elder [20] address a series of fundamental interrogations.

**Table 4.** Intellectual standards explained.

| Intellectual Standards | Tools for Assessing |
|---|---|
| Clarity | Could you elaborate further? Could you give me an example? Could you illustrate what you mean? |
| Accuracy | How could we check on that? How could we find out if that is true? How could we verify or test that? |
| Precision | Could you be more specific? Could you give me more details? Could you be more exact? |

**Table 4.** *Cont.*

| Intellectual Standards | Tools for Assessing |
|---|---|
| Relevance | How does that relate to the problem?<br>How does that bear on the question?<br>How does that help us with the issue? |
| Depth | What factors make this a difficult problem?<br>What are some of the complexities of this question?<br>What are some of the difficulties we need to deal with? |
| Breadth | Do we need to look at this from another perspective?<br>Do we need to consider another point of view?<br>Do we need to look at this in other ways? |
| Logic | Does all this make sense together?<br>Does your first paragraph fit in with your last?<br>Does what you say follow from the evidence? |
| Significance | Is this the most important problem to consider?<br>Is this the central idea to focus on?<br>Which of these facts are most important? |
| Fairness | Do I have any vested interest in this issue?<br>Am I sympathetically representing the viewpoints of others? |

Recent studies suggest that the development of CT skills is expected to be observed within one semester from the moment of implementing a dedicated curriculum [21]. Nevertheless, a more extensive observation is required. For that reason, the present study measured CT over the entire academic year of 2021–2022.

The aim of the present research article could be defined by the following main research questions:

- To explain how CT could be implemented into university curricula.
- To ascertain the students' viewpoints on the newly implemented CT blended apprenticeship curricula.

## 3. Methodology and Procedures

The present research is part of the international Erasmus project "Critical Thinking for Successful Jobs". One of the objectives of the project is to develop 15 CT blended apprenticeship curricula for the disciplines addressed by the consortium.

The methodology employs quantitative and qualitative methods of research. The theoretical method of research included the relevant scientific literature on CT and its importance in the educational field. The empirical method consists of a survey with closed/open questions. CT self-assessment (CTAS) and CT dispositions (CTDS) were tested. The survey was conducted by Vilnius University professors, for the academic year 2021–2022. The survey was based on an online questionnaire designed by one of the partners of the consortium, the University of Evora.

The CT blended apprenticeship curriculum tested was based on updated course descriptions that incorporated CT.

To assess the efficiency of the newly implemented CT blended apprenticeship curriculum in the discipline of Foreign Language Teaching, Vilnius University's students were asked to respond to a questionnaire before, during, and after the implementation of the curriculum (pre-questionnaire, mid-questionnaire, and post-questionnaire).

The questionnaire, designed to test the freshly implemented CT blended apprenticeship curriculum, consists of a set of 81 closed questions, organised in 2 different scales: the Critical Thinking Self-Assessment Scale (CTSAS) and The Student-Educator Negotiated Critical Thinking Dispositions Scale (SENCTDS).

Assessing CT skills and dispositions after implementing a specific curriculum designed to improve CT is one of the major challenges in higher education. Although there is a

multitude of theoretical tools proposed by researchers [22], the present study applies CTSAS and SENCTDS. The use of standardised tests does not fit the purpose of the present study due to their limitation in terms of applicability. Carreira R.P et al. [23] argue that standardised tests, in addition to not working in interdisciplinary settings, require specific expertise regarding data evaluation and interpretation. On the other hand, the multiple-choice responses severely limit the options of the responder.

The use of the SENCTDS is intended to increase the involvement of students in designing or constantly readjusting the curriculum. Quinn S. et al. [24] suggest that SENCTDS is a tool that reflects better the collaboration between student–teacher, an essential factor in assessing CT dispositions in higher education institutions. One of the methodological limitations (difficult to mitigate) of SENCTDS might be the apparent overlap of dispositions.

The respondents are students of the BA study programme of International Relations and Political Science. The survey was conducted for the academic year of 2021–2022 (between September and May) We collected data from 30 respondents for the pre-test, 31 respondents for the mid-test, and 29 for the post-test. The sample is relevant for the total number of students who actively participated in the activities and tasks proposed by the blended apprenticeship curriculum.

All respondents are 1st-year students, aged 19 to 21. English for Academic Purposes and Research is a two-semester subject, with 2 classes per week, 64 contact hours per semester, and 71 self-study hours.

The pre-test was conducted on 7 October 2021, two weeks after the beginning of the academic year. The mid-test was conducted on 8, 9, and 10 March 2022. The post-test was conducted on 10 May 2022.

The present research considers a predetermined methodological limitation. When testing the blended apprenticeship curriculum, we acknowledge the impact of other untested variables on the development of participants' CT. Students might have developed CT skills as a result of attending other courses or as a consequence of various social factors that could not be correlated with the tested curriculum. Nevertheless, the research still remains relevant, since English for Academic Purposes and Research is the only subject the participants have studied in the English language and the sole subject where CT skills and dispositions are mentioned explicitly and integrated coherently into the curriculum.

## 4. Description of the CT Blended Apprenticeship Curriculum

Our approach considers the constant tendency to implement challenge-based learning in higher education worldwide [25] and to enhance CT through active learning methods [26].

The overall doctrine of the curriculum is based on the task-based and action-oriented approaches, a system implemented by Vilnius University in Foreign Language Teaching starting with the academic year of 2019–2020. The tasks designed for the students are intended to be authentic, employing problem-solving and real-life situations. In designing the curricula, we considered the findings of several studies. Bezanilla et al. [27] argue that from the teachers' perspective, lectures are the least effective methodology for teaching CT in universities.

In the centre of the curriculum lies a scenario that replicates authentic cases. Students play a concrete role (political advisers, political analysts, etc.), and they are asked to provide solutions for the problems described. When trying to formulate solutions or make recommendations, students make proper use of the theoretical frameworks previously analysed in class. The theoretical frameworks include reading and discussing relevant research articles, defining and understanding specific concepts, and becoming familiar with the requirements of academic writing and research. In addressing the problem, students are supposed to use CT skills and dispositions, explicitly defined and properly explained to them, in line with the theoretical findings of Facione [18] and Paul and Elder [19].

The curriculum includes two forms of public speaking. In the first semester, students take part in parliamentary debates (based on the British format of the parliamentary

debate). They represent the government (Prime Minister and Deputy Prime Minister) and the opposition (Leader of the Opposition and the Deputy Leader of the Opposition) and provide various arguments in favour or against motions meant to discuss relevant topics in their fields of study. Students who are not debating are asked to provide their feedback and comments. Argumentation and feedback contribute to the questioning of one's beliefs and knowledge [28]. The role of the debates in cultivating CT was explored by Chen et al. [29]. The study suggests that students should be encouraged to play an active role in the debates by various methods, such as rearranging the space or recording the entire activity.

In the second semester, the public speaking activity includes an individual argument. Each student addresses an authentic real-life situation, taking the role of a decision-making persona. The task is organised as a press conference: at the end of the individual argument, the speaker should answer questions from their colleagues who play the role of journalists. In terms of developing CT skills, the main goal is to engage students in a rational argument and to enhance their ability to make the distinction between an opinion and an argument. The authentic scenarios, where students try to provide solutions for daily life problems in their future field of expertise, are meant to motivate them. Between motivation and the enhancement of CT skills and dispositions, there is a correlation [30] that our curricula propose to address.

Students are meant to familiarise themselves with the requirements of academic writing. For this purpose, the curriculum proposes two different tasks. Firstly, they acquire theoretical knowledge about the valid structure of scientific research. After having a proper understanding of the genre, they are asked to submit a research proposal, meant to prepare and anticipate the next task. Secondly, at the end of the first semester, we simulate an international conference where each student is expected to contribute with an individual presentation. The same activity is organised at the end of the 2nd semester, only this time students work in teams. The activity is structured as an open event, where other students or teachers could participate. Both presentations are followed by a Q&A session, where the presenters respond to questions, comments, or suggestions addressed by the audience. The tasks start from a well-defined, authentic scenario, based on the general topics mentioned in the course description. In terms of CT, the tasks test the intellectual traits mentioned above [19].

## 5. Results and Discussion

Although the CTSAS shows only minor progress between the pre-test and the mid-test, the post-test clearly indicates a significant improvement in terms of CT. The percentage of respondents who replied with "usually/often/frequently/always" notably increased for 17 of the questions addressed in CTSAS. A moderate increase was registered for 43 questions (Figure 1).

For the following 17 questions, the survey indicated a positive difference between the pre-test and post-test:

When presented with a problem:

I classify data using a framework.
I examine similarities and differences among the opinions posed for a given problem.
I examine the inter-relationships among concepts and opinions posed.
I ask questions in order to seek evidence to support or refute the author's claim.
I figure out if the author's arguments include both for and against the claim.
I figure out unstated assumptions in one's reasoning for claim.
I look for the overall structure of the argument.
I figure out the assumption implicit in the author's reasoning.
I collect evidence supporting the availability of information to back up opinions.
I systematically analyse the problem using multiple sources of information to draw inferences.
I analyse my thinking before jumping to conclusions.

It could be argued that the scale measured the perception; nevertheless, it is undeniable that the results show a significant improvement in terms of awareness.

Considering the results, we could safely ascertain that the participants are more inclined to consider applying CT techniques when tackling various problems. Another interesting result indicates that participants developed the habit of questioning statements and standpoints that often present themselves as axiomatic.

In terms of intellectual traits, the comparison of the results of the pre-test and post-test demonstrated the presence of intellectual humility. If the pre-test revealed the propensity for considering certain conjectures as unquestionable, the post-test suggested a different approach, higher flexibility, and a desire to closely scrutinise various points of view (Figure 2). Asked whether they will review their reasons and reasoning process in coming to a given conclusion, participants showed in the post-test a higher predilection to do so (16.66%, 35.48%, and 41.37%).

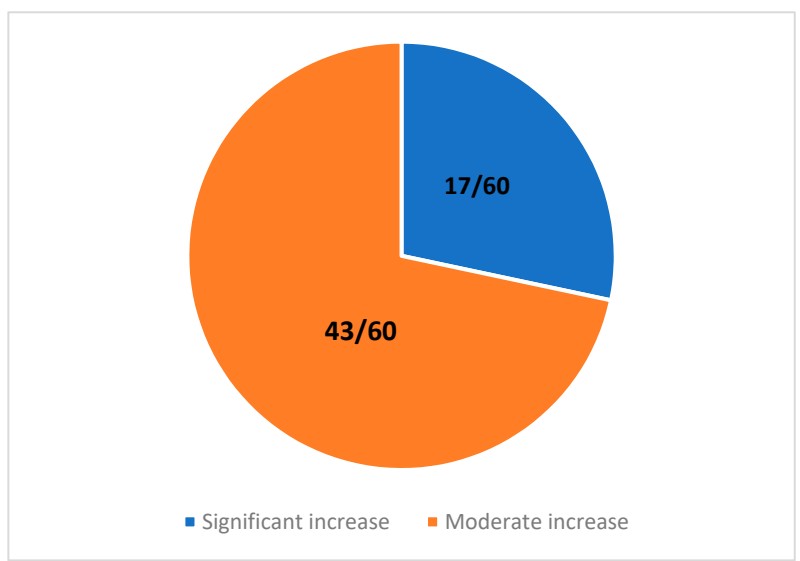

**Figure 1.** Comparative analysis. General progress report. "Usually/often/frequently/always" responses to CTSAS out of 60 questions.

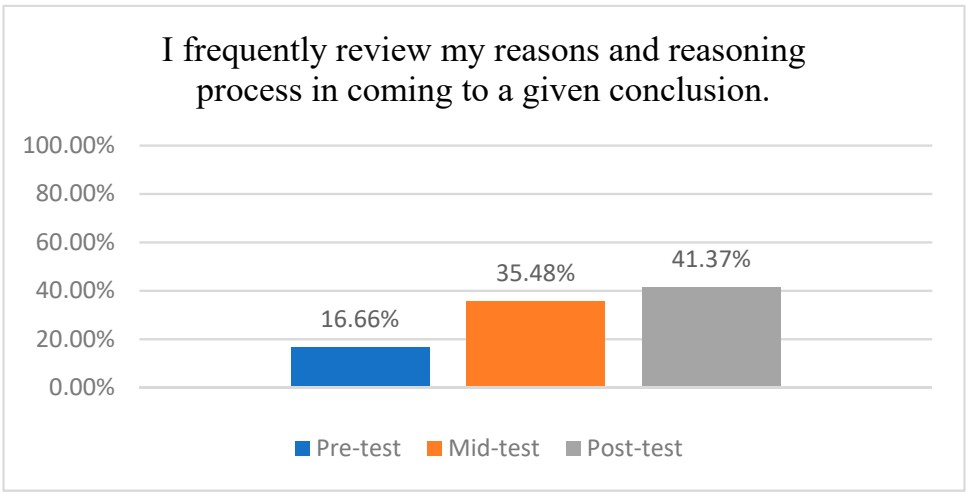

**Figure 2.** Indicators of intellectual humility.

An interesting observation refers to the weight students give to non-verbal communication in class (Figure 3). The results indicate that, from pre-test to post-test, the number of participants who carefully observe the facial expression people use in a given situation

remains high. The fluctuations between the pre-test and post-test remain minimal, while the mid-test data indicate a decrease. In terms of CT, the observation should be correlated with other indicators for a more substantial conclusion. Nevertheless, the importance of non-verbal communication has to be mentioned as a challenging factor.

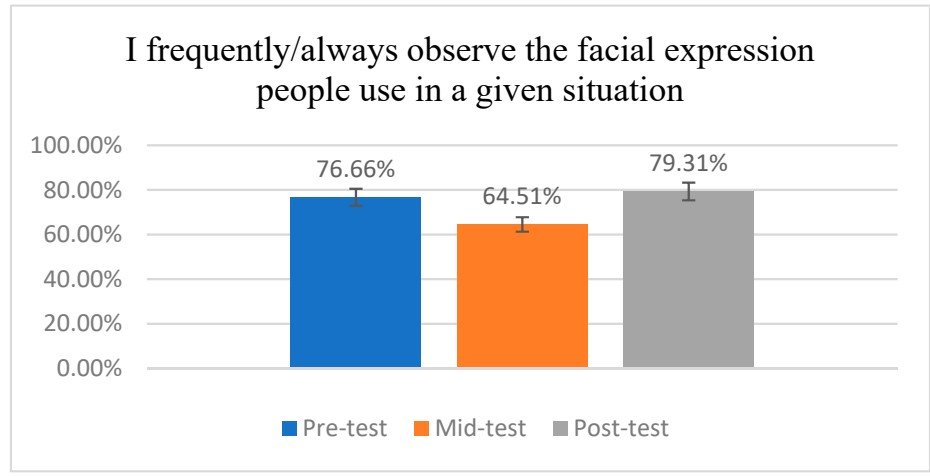

**Figure 3.** The importance of non-verbal communication.

Facial expressions are culturally grounded; decoding a facial expression implies being familiar with the cultural code of the emitter of the message. On the other hand, the preoccupation for analysing and understanding facial expressions might correlate with one of the intellectual traits defined by Paul and Elder, intellectual empathy.

In Figure 4, we enlisted the most significant progress registered from the pre-test to the post-test. The data collected indicate that the blended apprenticeship curriculum tested enhanced analytical awareness. Participants prove an augmented predisposition towards higher-order thinking skills.

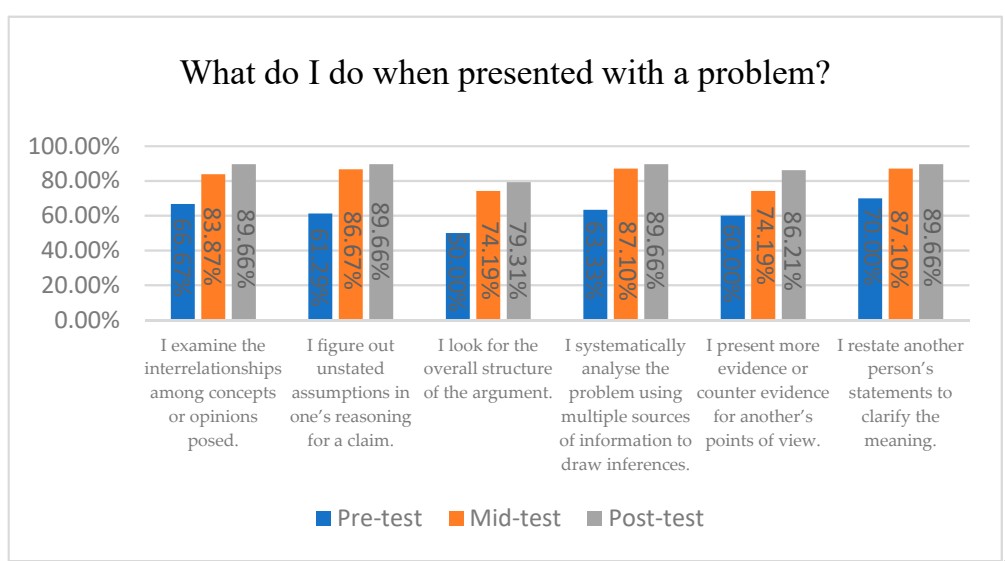

**Figure 4.** Progress report CTSAS (top 6). "Usually/often/frequently/always" responses.

Only 50% of the participants in the pre-test declared that, when presented with a problem, they looked for the overall structure of the argument. The percentage increased to 74.19 in the mid-test and reached 79.3 in the post-test (Figure 4). The results correlated with one of the major aims of the blended apprenticeship curriculum: the distinction between an argument and an opinion. Students were presented with the structure of the argument

and were encouraged to decide on the validity of a statement based on the arguments provided for or against it. The parliamentary debates (first semester) and the individual argument (in the second semester) were tasks specially designed to address the importance of a solid argumentation. The data suggest that public speaking can effectively foster CT skills and predispositions.

The tested blended apprenticeship curriculum included various tasks (individual presentation, team presentation, and research proposal) that aimed at enhancing the students' understanding of theoretical concepts relevant to their field of study. The proper use of terminology is more often than not overlooked, resulting in terminological confusion or even functional illiteracy. At the beginning of the academic year, only 66.67% of the participants examined the inter-relationships among concepts or opinions posed when presented with a problem. The result of the post-test indicates an increase of 2.99%.

On the other hand, SENCTDS indicates that the CT dispositions were not significantly improved by the tested blended curriculum. Where the progress could be observed, it is still inconsequential (Figures 5 and 6). In some cases, even a decrease was observed. It is partly explained by subjective unpredictable conditions. The post-test was conducted towards the end of the academic year, when usually stress and the ability to focus on certain tasks deteriorate. To mitigate the risk, we might consider a better assessment strategy. The competitiveness increases in the second semester, when students receive marks instead of pass/fail assignments (first semester). Traditionally, marks constitute a stimulus but might put extra pressure on students, with visible consequences on their willingness to enforce and make use of their CT dispositions. The hypothesis is not tested, and we do not have enough data to support it.

The data show that, at the beginning of the academic year, 60.33% of the participants enjoyed dealing with information that aroused their curiosity, even if the material was difficult. Towards the end of the academic year, the figure dropped to 51.61% and then 55.17%. A similar result indicates a deterioration in the enthusiasm for learning new things. The pre-test indicated that 60.00% of the participants enjoyed the information that challenged them to think. The figure slightly dropped for the mid-test and the post-test.

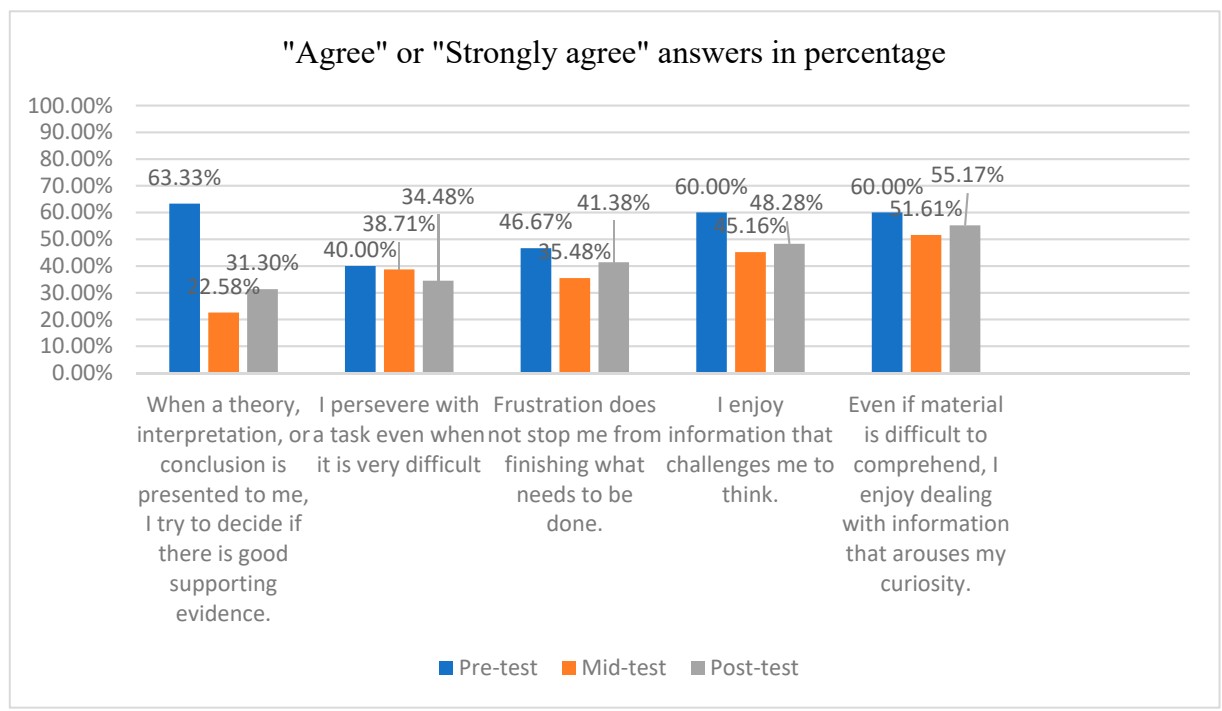

**Figure 5.** SENCTDS. Evolution of CT dispositions.

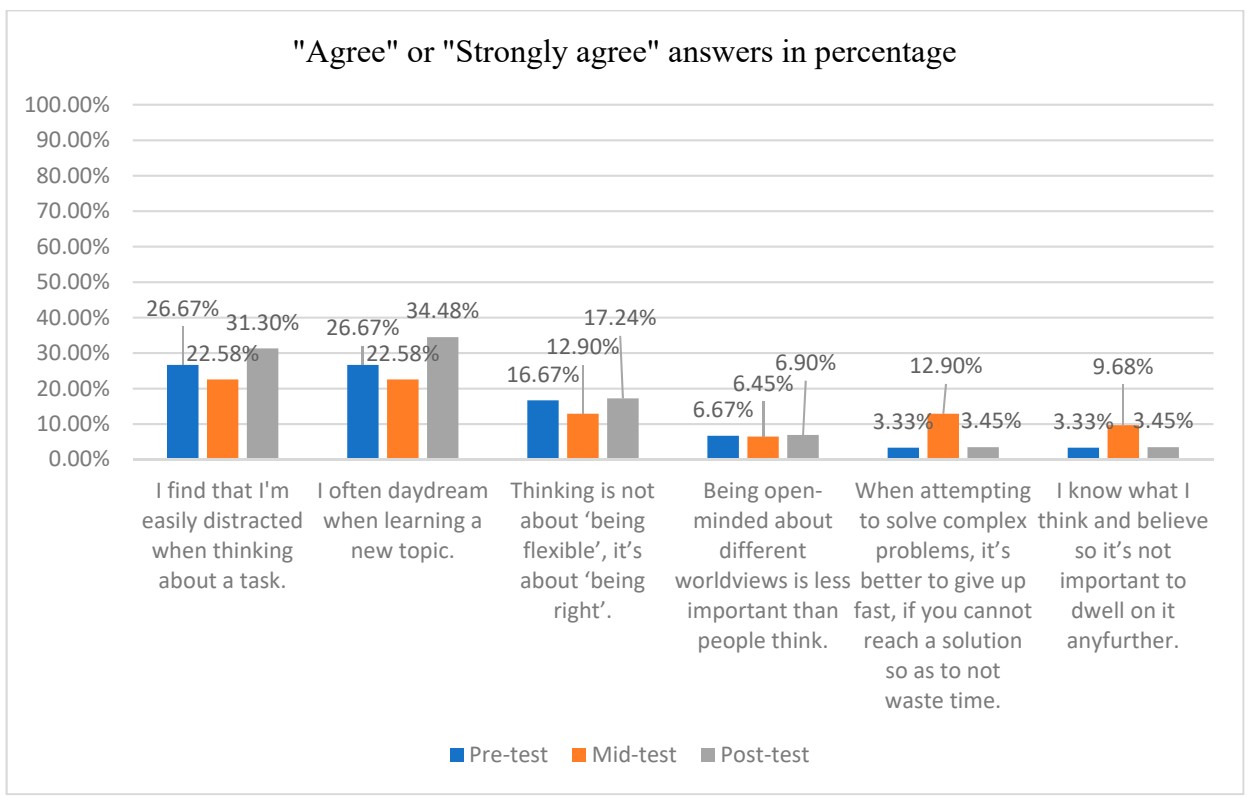

**Figure 6.** SENCTDS. The decline in CT dispositions.

The results presented in Figures 5 and 6 are consistent with the hypothesis formulated above. Fatigue and increasing competitiveness seemed to have a negative impact on CT dispositions. A better redistribution of tasks and an assessment strategy aligned with the results of the present study might address the problem. However, a proper conclusion requires expanding the research and collecting data from other study programmes. Only correlating data from a larger sample and conducting surveys for a longer time might provide us with a broader picture.

## 6. Conclusions

The results collected (SENCTDS) indicate that CT skills are more susceptible to improvement in the short term than CT dispositions. An improvement in CT skills could be observed in a fairly decent interval (one semester).

CT dispositions depend on multiple other variables difficult to quantify and assess. They are often connected to core beliefs and cultural values resistant to change [31]. The data collected imply that CT dispositions slightly deteriorated towards the end of the academic year. The findings appear to suggest that the tested curriculum requires a better redistribution of tasks throughout the two semesters.

Overall, the blended apprenticeship curriculum contributed to improving CT. For most of the skills tested by CTSAS, data suggest significant progress. The authentic scenario, incorporating a clearly defined target audience and the role of the speaker, contributed to stimulating the learning environment. The core philosophy of the action-oriented approach is to provide the students with authentic situations (scenarios) where their knowledge can be used pragmatically. The shift from the abstract, theoretical approach to a realistic approach resulted in a tangible increase in CT skills. Participants have a more accurate understanding on how and why they should apply different methods of analysis. They are able to see their knowledge at work in various real-life cases.

**Author Contributions:** Conceptualization, O.I.; methodology, O.I.; software, O.I.; validation, O.I., R.K. and S.P.; formal analysis, O.I.; investigation, O.I.; resources, O.I., R.K. and S.P.; data curation, O.I.; writing—original draft preparation, O.I.; writing—review and editing, O.I., R.K. and S.P.; visualization, O.I.; supervision, O.I.; project administration, R.K.; funding, R.K. All authors have read and agreed to the published version of the manuscript.

**Funding:** This work has been supported by the "Critical Thinking for Successful Jobs—Think4Jobs" Project, with the reference number 2020-1-EL01-KA203078797, funded by the European Commission/EACEA, through the ERASMUS + Programme. The European Commission support for the production of this publication does not constitute an endorsement of the contents, which reflect the views only of the authors, and the Commission cannot be held responsible for any use which may be made of the information contained therein.

**Institutional Review Board Statement:** The animal study protocol was approved by the Institutional Review Board of the Vilnius University Faculty of Philology (No. 180000-S-224 and 8 December 2022).

**Informed Consent Statement:** Informed consent was obtained from all subjects involved in the study.

**Data Availability Statement:** Additional data about research can be found in Intellectual Output 3, available at: https://think4jobs.uowm.gr/wp-content/uploads/2022/03/EN_IO3_Think4 Jobs_ebook.pdf (accessed on 15 February 2023) and the website of the project at: https://think4jobs. uowm.gr/ (accessed on 15 February 2023).

**Conflicts of Interest:** The authors declare no conflict of interest.

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
