# Peer review of "Implementation of the Critical Thinking Blended Apprenticeship Curricula and Findings per Discipline: Foreign Language Teaching"

_education, doi:10.3390/educsci13020208_

Round 1

Reviewer 1 Report

Dear Authors, Thanks for your interesting study. There are some points we like to comment on a) the used literature considering CT as concept and on didactical aspects, b). on the conclusions and the empirical design.

a): CT as a concept: there are some crucial aspects highlighted considering the concept of CT (Chapter 1 & 2) in your text, but the great body of work/literarture from America with regard to CT is missing, although you use many concepts from there (Paul and Elder, Facione etc.). Moreover, there is hardly literature used with regard to fostering CT, although there are even comprehensive meta-studies on fostering CT available (for example Abrami et al., 2015, 2008). In these metastudies you can find also the effects that an introduction of CT in curricula achieve in general (Abrami et. al , 2008), for example.

b): Empirical design:

- Test: Why did you use the CTSAS & SENCTDS-tools and not other established tests and instruments? What are the strengths and the weaknesses of that questionaire?  Here a methodological reflection seems neccessary.

- Description of the results: CTSAS-question and the increase should be explained a little bit more e. g. What kind of questions are asked in the "significant increase"-case (Figure 1)? Why did you pick the presented questions? Figure 4: Items are not fully readable etc.

- Conclusions: Some unchecked hypothesis seem worthy for consideration like: maybe the introduction of marks puts the focus on the reproduction of knowledge given by the teacher (say or write the "right" answer) and not on the quality of thinking and it's progress. Hence, this may also alter the teaching styles (knowledgetransfer in stead of promoting CT).

- Practical impact: For practitioners some more notes on didactical designs could be helpful (for example: how are the parliamentary debates are carried out? what is important in the process? etc.)     

Sincerely,

Hannah Fontana Reviews ;-)

Author Response

Dear Reviewer 1,

Thank you for carefully reading our manuscript and providing interesting observations and insight. We will address your comments as follows:

  • We included relevant literature regarding fostering CT: Abrami et al. (2014). The manuscript now also includes observations on course and curriculum content (Abrami et al., 2008).
  • Following your observations on the strengths and weaknesses of CTSAS & SENCTDS tools, we included the results of two studies. Carreira R.P et al., “Development and Validation of a Critical Thinking Assessment-Scale Short Form”, argue that the shortening of CTSAS (Nair) removed the redundant items and eliminated those items that did not focus on cognitive skills. The aforementioned research extensively explains the modifications done and the reasoning (chapter 2, 4-9). The same study addresses the weaknesses of standardised tests: standardised tests do not function properly across disciplines, and they pose several difficulties in terms of functionality (difficult and expensive to apply). The second study - Quinn S. et al., “Development and validation of the Student-Educator Negotiated Critical Thinking Disposition Scales” (2020). One of the weaknesses might be the apparent disposition overlap (Quinn et al.). Further explanations were provided in the updated version of the manuscript.
  • To address your observation, we included more details about the data in the manuscript in figure 1. Figure 4 was modified; the results should be readable now.

Reviewer 2 Report

The paper is very interesting and aims to analyse how Critical Thinking can be measured and tested in higher education. Furthermore, it proposes specific tasks designed to increase the use of Critical Thinking. However, there are some critical points to be reconsidered:

1.     Critical Thinking Categories in Tables 1 to 4 are presented without an effort to explain a possible relation between them. If there is any connection between these tables, it is not apparent in the Introduction.

2.     Why do the authors refer to research tasks and not to research questions? I think that with slight changes, the research tasks could be presented as research questions.

3.     I would propose the results be presented in a scale-based approach instead of an item-based approach (for instance, figures 4, 5 and 6) in favour of robustness. For instance, the authors could use the validated assessment CTSAS scale (Payan-Carreira et al., in the same special issue, https://doi.org/10.3390/educsci12120938) to present progress in this scale (Fig. 4). On the other hand, progress in SENCTDS scale could be presented in a scale-based approach deploying a-Cronbachs’ indexes/results.

There are also some second-level comments:

4.     CT in the title should be Critical Thinking; acronyms are not preferred in a title.

5. In-text references do not follow the journal's rules/instructions.

6.     Dates in lines 79 and 84 seem to mismatch. Please, check the years that are referred to in those lines.

7.     Table 1, Interpretation, I assume you meant “clarifying” instead of “clarigying”.

8.     Table 3 should be correctly enumerated.

9.     Table 3, Confidence in Reason, has no Descriptor, shouldn’t there be one?

Author Response

Dear Reviewer 2,

Thank you very much for carefully reading our manuscript and for providing interesting observations and insight. We will definitely address your observations as follows:

We added one more phrase meant to clarify the connection between table 1 and table 4.

We acknowledge that the subtle terminological distinction between “research tasks” and “research questions” does not constitute the purpose of the present study. Consequently, following your suggestions, we replaced “research tasks” with “research questions”.

We replaced the acronym in the title.

We modified the in-text references according to the requirement of the journal.

The ambiguity between line 79 and line 84 was removed.

The spelling mistake identified in table 1 was removed.

Table 3 was now correctly enumerated.

In table 3, we added a descriptor for “confidence in reason”.

Reviewer 3 Report

Although the article is not particularly original, its value lies in its authenticity. The author(s) of the article use classical criteria for assessing critical thinking skills and dispositions, and apply popular schemes to a specific students' audience. 

The aim of the article is clear, as are the research objectives and methodology. The results of the study are presented appropriately, the conclusions are logical and relevant. The language used is professional and fluent. 

It would be useful to supplement the discussion with a comparison with other similar studies in HE of this kind. Especially as P. Facione's methodology is widely used   to assess the development of critical thinking skills and dispositions. After this small improvement the article may be accepted for publishing.

Author Response

Dear Reviewer 3,

Thank you very much for carefully reading our manuscript and for providing interesting observations and insight. We will definitely address your observations as follows:

Following your suggestion to make a comparison with other similar studies in HE, we expanded our bibliography and incorporated into our manuscript references to other studies conducted in the field. Below, we mention the research that was added:

Abrami, P.C., Bernard, R.M., Borokhovski, E., Waddington, D.I., Wade, C.A., Persson, T.: Strategies for teaching students to think critically: a meta-analysis. Review of Educational Research. 85(2), 275-314 (2014). DOI: 10.3102/0034654314551063

Abrami, P.C, Bernard, R.M., Borokhovski, E., Wade, A., Surkes, M.A., Tamim, R., Zhang, D.: Instructional interventions affecting Critical Thinking skills and dispositions: a stage 1 meta-analysis. Review of Educational Research. 78(4), 1102-1134 (2008). DOI: 10.3102/0034654308326084

Quinn, S., Hogan, M., Dwyer, C., Finn, P., Fogarty, E.,: Development and Validation of the Student-Educator Negociated Critical Thinking Dispositions Scale (SENCTDS). Thinking Skills and Creativity. 38 (2020). doi.org/10.1016/j.tsc.2020.100710

Payan-Carreira, R., Sacau-Fontenla, A., Rebelo, H., Sebastiao, L., Pnevmatikos, D., Development and validation of a Critical Thinking Assessement-Scale Short Form. Critical Thinking: Bridging a Successful Transition between University and Labour Market. 12 (2022). doi.org/10.3390/educsci12120938

Liu, O.L., Frankel, L.,Roohr, K.C., Assessing Critical Thinking in Higher Education: Current State and Directions for Next-Generation Assessements. ETS Research Report Series. 1, 1-23 (2014). doi.org/10.1002/ets2.12009